# Hardware Emulation of Step-Down Converter Power Stages for Digital Control Design

Botond Sandor Kirei [1], Calin-Adrian Farcas [1], Cosmin Chira [2], Ionut-Alin Ilie [2] and Marius Neag [1,*]

1   Department of Bases of Electronics, Technical University of Cluj-Napoca, 400114 Cluj-Napoca, Romania;
    botond.kirei@bel.utcluj.ro (B.S.K.)
2   Infineon Technologies, 020335 Bucharest, Romania
*   Correspondence: marius.neag@bel.utcluj.ro

**Abstract:** This paper proposes a methodology of delivering the emulation hardware of several step-down converter power stages. The generalized emulator design methodology follows these steps: first, the power stage is described using an ordinary differential equation system; second, the ordinary differential equation system is solved using Euler's method, and thus an accurate time-domain model is obtained; next, this time-domain model can be described using either general-purpose programming language (MATLAB, C, etc.) or hardware description language (VHDL, Verilog, etc.). As a result, the emulator has been created; validation of the emulator may be carried out by comparing it to SPICE transient simulations. Finally, the validated emulator can be implemented on the preferred target technology, either in a general-purpose processor or a field programmable gate array. As the emulator relies on the ordinary differential equation system of the power stage, it has better behavioral accuracy than the emulators based on average state space models. Moreover, this paper also presents the design methodology of a manually tuned proportional–integrative–derivative controller deployed on a field programmable gate array.

**Keywords:** buck converter power stage; ordinary differential equation system; hardware emulation; field programmable gate arrays





## 1. Introduction

Lately, the digital implementation of the control loops in switched-mode power supplies (SMPS) gains larger and larger attention because digital technology presents certain advantages over its analog counterpart: design flexibility through programmability, lower sensibility to component tolerances, no need for passive tuning components, etc. [1]. Usually, the simulation of digital and analog sections of a buck converter is carried out in a mixed-signal simulation environment, as the power stage is regularly modeled as an electric circuit. At the same time, the control loop is described in general-purpose programming language or hardware description language. An accurate time-domain model for the buck converter is developed to reduce the gap between the analog and digital sections. The model can be simulated in any numerical computation environment, and in a traditional, event-driven logic simulator [2]. The model can be synthesized targeting field programmable gate arrays (FGPA); thus, a power stage hardware emulator can be devised. Therefore, there is no need for the actual power stage in the digital control development, a situation fitted for a high-power converter control loop design [3].

Ref. [4] emphasizes the importance of power stage simulation in a proper event-driven simulation environment, as it can potentially aid the development of chip integrated SMPS. In [4], a SystemVerilog model of a boost power stage is achieved. In their modeling approach, the differential equations of the power stage in both operation phases are composed, and their Laplace transform is achieved and finally solved as described in [5]. This modeling approach achieves SPICE-level accuracy with 20~100× faster simulation

speeds. A significant downside of the event-driven simulation methodology in [4] is that the proposed SystemVerilog model is not synthesizable; thus, it can be used only in functional simulations, as hardware emulation is not possible. Another disadvantage is the modeling accuracy; in the exemplified boost power stage model no parasitic components were considered.

Ref. [6] describes power electronics device emulation hardware in FPGA. In their approach, an insulated gate bipolar transistor (IGBT) and the power diode are emulated based on Hafner [7] and Lauritzen [8] models, respectively. Using these two devices, a buck converter was emulated. An advantage of [6] is that it emphasizes the dynamic behavior of the switching devices. In contrast, the proposed hardware emulation focuses on the behavior of the switching converter as an overall system, grasping the two most important values: the inductor current and capacitor voltage.

Several papers highlight the significance of real time simulation (RTS)—in other words, hardware emulation—for hardware in the loop (HIL)-enabled development methodologies. Ref. [9] deals with the RTS of a phase shifter converter for high frequency applications. It demonstrates the necessity of a small simulation step—less than 20 ns for a 200 kHz SMPS—for avoiding false limit cycling behavior and obtaining an accurate closed loop response by the RTS. Thus, the natural choice to implement an RTS was a FPGA which confers a necessary bandwidth for the computations. Ref. [10] tackles HIL technology aiming control loop development for a buck converter. The shortcoming of their achievement is a reduced accuracy, due to the use of averaged models running on a general-purpose processor with limited computational power. A large power system simulation in FPGA is addressed in [11]. The most widespread solution to simulate/emulate a large system is to separate it into subsystems. Unfortunately, partitioning may introduce simulation time step latency between different subsystems, which may cause numeric instabilities. This situation is solved in [11] by applying a predictor/corrector numerical integration method (a combination of forward and backward Euler solver). The aforementioned emulators [9–11] are, in essence, custom, low-cost solutions. Professional tools are also available, such as eFPGASIM [12], a powerful and intuitive FPGA-based emulator.

The original contributions of this work to the state-of-the-art solutions are summarized as follows:

- A generalized methodology for FPGA-based power stage emulation is formulated. The methodology proposes that the hardware emulator shall imitate the capacitor voltage and the inductor current or currents in the case of a multiphase topologies, the two most interesting values used in a voltage/current mode control loop [13];
- The proposed methodology devises the following steps: first, the power stage is described using an ordinary differential equation (ODE) system. Second, a numerical integration method [14] is used, such as 1st order forward Euler, 2nd order Adams–Bashforth, 2nd order Runge–Kutta, and 4th order Runge–Kutta [15], to formulate an iterative process that solves the ODE system. Thus, an accurate time-domain model is obtained. Next, the resulting iterative process is implemented using either general-purpose programming language (MATLAB, C, etc.) or hardware description languages (VHDL, Verilog, etc.), depending on the desired emulator target technology. As an optional step, we recommend the validation of the iterative process. This can be carried out by a comparison with a transient simulation using a SPICE-like simulator. Finally, the validated iterative process can be implemented on the preferred target technology, either in a general-purpose processor or in dedicated hardware, such as an FPGA;
- The demo of the proposed methodology for several buck power stage topologies is given in this paper: (i) ideal synchronous buck, (ii) synchronous buck converter with parasitic components (direct current resistance of the inductance, the equivalent series resistance of the capacitor, on-resistance of MOSFET switches), and (iii) ideal asynchronous buck. These topologies were emulated using a hardware description

language because FPGAs are the de-facto choice for an emulator target technology, as it can offer simulation steps at the order of nanoseconds [16];

- VHSIC Hardware Description Language (VHDL) was used to create synthesizable hardware and code snippets are also given in the paper;
- Our solution is compared with existing ones, given in refs. [9–12];
- A digital proportional–integrative–derivative (PID) controller design method was devised using emulated hardware (real-time simulation). The hardware emulation allows the designer to manually tune the PID controller coefficients, without any harm to the actual power stage. After the controller was tuned, the digital PID control loop was connected to the actual power stage and comparison with conventional voltage mode control was carried out.

The paper is organized as follows: in Section 2, a step-by-step guide for creating a synthesizable hardware description of several buck power stage topologies (synchronous, asynchronous, and synchronous with parasitic components) is given. The developed VHDL models are validated through comparison: the estimated capacitor voltage and the inductor currents are compared to an electrical circuit model simulated in LTSPICE. Although the VHDL is synthesizable, pipelining and numerical optimization is advised, as foreseen in Section 3. In Section 4, a manually tuned PID controller design methodology is presented. In Section 5, the results are revisited and commented on. Finally, conclusions are drawn in the last section.

## 2. Hardware Emulator Development Methodology

In this section the hardware emulator design methodology is presented. The starting point is the buck converter topology, while the output is a synthesizable VHDL code that can be deployed on FPGA. In Figure 1, the synthesizable VHDL model development and validation methodology applied in this paper is presented. To establish the VHDL model, the power stage's ODEs are formulated. In general, the symbolic solution of an ODE system is challenging to obtain, but sometimes an approximate numerical solution is enough. Several numerical methods for solving ODEs are available. One is the very well-known Euler's method [17]. The numerical solution of the ODE can be traced using an iterative process. The iterative process can be implemented in general-purpose programming languages (MATLAB, C, etc.) or hardware description languages (VHDL, Verilog, etc.). The FPGA is the target technology we choose in this paper, so we created VHDL modules to implement the iterative process.

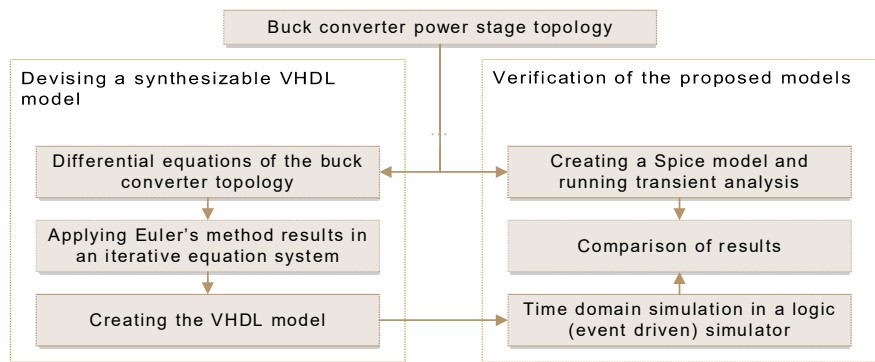

**Figure 1.** Synthesizable VHDL model development and validation methodology.

The validation of the VHDL model is carried out by comparing the capacitor voltage and inductor current waveforms resulting from an event-driven (logic) simulator to their counterparts extracted from a transient analysis of the given topology carried out in LTSPICE. As we will show later, the differences between the two simulations are negligible.

In the following, we guide the reader through the creation of three VHDL modules for: ideal synchronous buck (no parasitic components are considered), ideal asynchronous

buck (the diodes nonideality factor, series resistance, and forward voltage are modeled), and a real(istic) buck converter (considering the on-resistance of the MOSFET switches, equivalent series resistance of the inductor and the capacitors).

*2.1. Ideal Synchronous Buck*

2.1.1. Ordinary Differential Equation System Derived from the Circuit Topology

The ideal synchronous buck converter model is depicted in Figure 2, alongside its operation modes. A driver circuit generates the pulse width modulated (PWM) signal noted $d(t)$. When $d(t) = 1$, then the controlled switch is on; otherwise, $d(t) = 0$, and the switch is off. In operation mode 1, the inductor stores the energy from the voltage source $V_{in}$, while in operation mode 2, the stored energy is transferred to the load. The differential equation systems for the equivalent circuit of Mode 1 and 2 are given in Equation (1), respectively (2).

$$\begin{cases} \frac{di_L(t)}{dt} = \frac{1}{L}\left(V_{in} - v_C(t)\right) \\ \frac{dv_C(t)}{dt} = \frac{1}{C}\left(i_L(t) - \frac{v_C(t)}{R}\right) \end{cases} \tag{1}$$

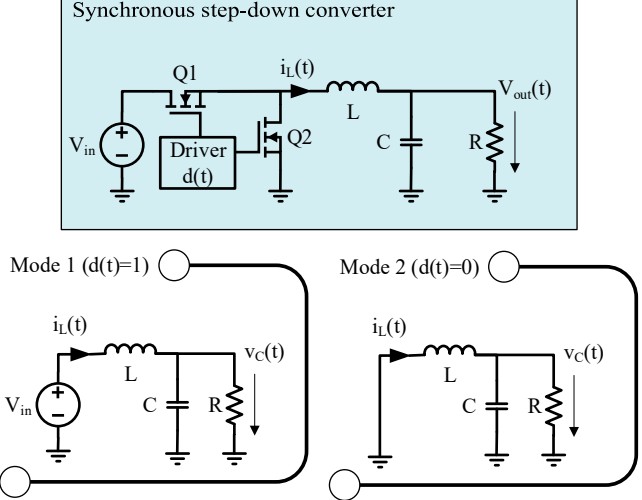

**Figure 2.** Ideal synchronous step-down (buck) converter model and its operating modes.

$$\begin{cases} \frac{di_L(t)}{dt} = -\frac{1}{L}v_C(t) \\ \frac{dv_C(t)}{dt} = \frac{1}{C}\left(i_L(t) - \frac{v_C(t)}{R}\right) \end{cases} \tag{2}$$

Equation systems (1) and (2) can be merged with the help of $d(t)$, which represents a PWM signal. Note that the value of $d(t)$ is 1 when the switch is on, and it is 0 when the switch is off. For the first and second modes, the change of the inductor current is expressed $d(t)*(V_{in} - v_C(t))/L$ and $-v_C(t)/L$, respectively. Overall, the inductor current variation is $d(t)*V_{in}/L + v_C(t)/L$.

Similarly, the variation of capacitor voltage $v_C$ can be obtained. The resulting ODE is:

$$\begin{cases} \frac{di_L(t)}{dt} = \frac{1}{L}\left[d(t)V_{in} - v_C(t)\right] \\ \frac{dv_C(t)}{dt} = \frac{1}{C}\left[i_L(t) - \frac{v_C(t)}{R}\right] \end{cases} \tag{3}$$

2.1.2. Numerical Integration with Euler's Method

The next step is to derive the numerical integration. The ODE in (3) is solved for the inductor current $i_L$ and the capacitor voltage $v_C$. Although $d(t)$ varies with time, it is treated as a parameter as its value is known over the simulated time interval. We also

assume $i_L(t_0) = 0$ *A* and $v_C(t_0) = 0$ *V* as initial conditions. The resulting iterative process and corresponding pseudocode are given in Equation (4) and Algorithm 1, respectively.

$$\begin{cases} i_L(t_0) = 0, v_C(t_0) = 0 \\ i_L(t_{n+1}) = i_L(t_n) + (d(t_n)V_{in} - v_C(t_n))\frac{\Delta t}{L} \\ v_C(t_{n+1}) = v_C(t_n) - \left(i_L(t_n) - \frac{v_C(t_n)}{R}\right)\frac{\Delta t}{C} \end{cases} \tag{4}$$

where $t_n = t_0 + n\,\Delta t$, *n* is the index of the sample, and $\Delta t$ is a sufficiently small value.

---

**Algorithm 1**: Ideal Synchronous Buck Model

---

**Input**: $\Delta t$, *L*, *C*, *R*, $V_{in}$, *d(t)*
**Output**: $V_{out}$, $i_L$
    1: **initialize** $V_{out} \leftarrow 0$, $i_L \leftarrow 0$
    2: **loop forever**
    3: **compute** $i_L$\_new $\leftarrow i_L + (d * V_{in} - v_C) * \Delta t/L$
    4: **compute** $v_C$\_new $\leftarrow v_C + (i_L - v_C/R) * \Delta t/C$
    5: **update** $i_L \leftarrow$ maximum($i_L$\_new, $1 \times 10^{-20}$)
    6: **update** $v_C \leftarrow v_C$\_new
    7: **end loop**

---

Algorithm 1 describes time domain simulation of inductor current $i_L(t)$ and capacitance voltage $v_C(t)$ in an ideal synchronous buck converter.

### 2.1.3. VHDL Implementation of Euler's Method

Algorithm 1 is suitable to be implemented in general-purpose programming language, such as MATLAB or C. Note that a C implementation could be run on a microcontroller platform to serve as a hardware emulator. It is up to the designer to use the emulated inductor current and capacitor voltage as a numerical value or undergo a digital to analog conversion.

### 2.1.4. Validation (Optional)

Algorithm 1 was implemented in VHDL. The actual VHDL code is attached in Appendix A. Although we consider that validation of the VHDL model is optional, some may find it useful to compare it with an electrical circuit simulation in a SPICE-like simulator. In this paper, the VHDL model was simulated in gHDL, an open-source event-driven logic simulator. The electrical model of an ideal buck converter was carried out and simulated in LTSPICE. The electrical circuit model is in Appendix B. The parameters of the emulated buck power stage are listed in Table 1.

**Table 1.** Buck power stage and simulation parameters.

| Parameters | Value |
| --- | --- |
| $\Delta t$ simulation step size | $10^{-8}$ s |
| Simulation time | 0.1 ms |
| *L*—inductor | 100 μH |
| *C*—capacitance | 1 μF |
| *R*—load | 10 Ω |
| $V_{in}$ | 10 V |
| Duty circle | 50% |
| Switching frequency | 1 MHz |

Simulations were carried out using both the VHDL description and electric circuit schematic, resulting in the waveforms in Figure 3. The capacitor voltages and inductor currents are illustrated in the first and the second plot, respectively. The differences are hardly distinguishable with the free eye. The direct comparison of the inductor currents and capacitor voltages is impossible, as the SPICE solver uses a variable step size for the time variable, while the VHDL functional model is a fixed step size model. Thus, the waveform samples were averaged in a sliding window, and their average values were compared in terms of relative error. In the third plot, the relative errors between voltage and current waveforms are presented. In the case of the voltage waveform, the peak relative error is approximately 10% in the transient state, but as the converter reaches steady-state operation the relative error is approximately 0.2%. The peak relative error of the current waveforms is approximately 10% (in the transient state). In a steady-state operation, the current relative error stays lower than 1%.

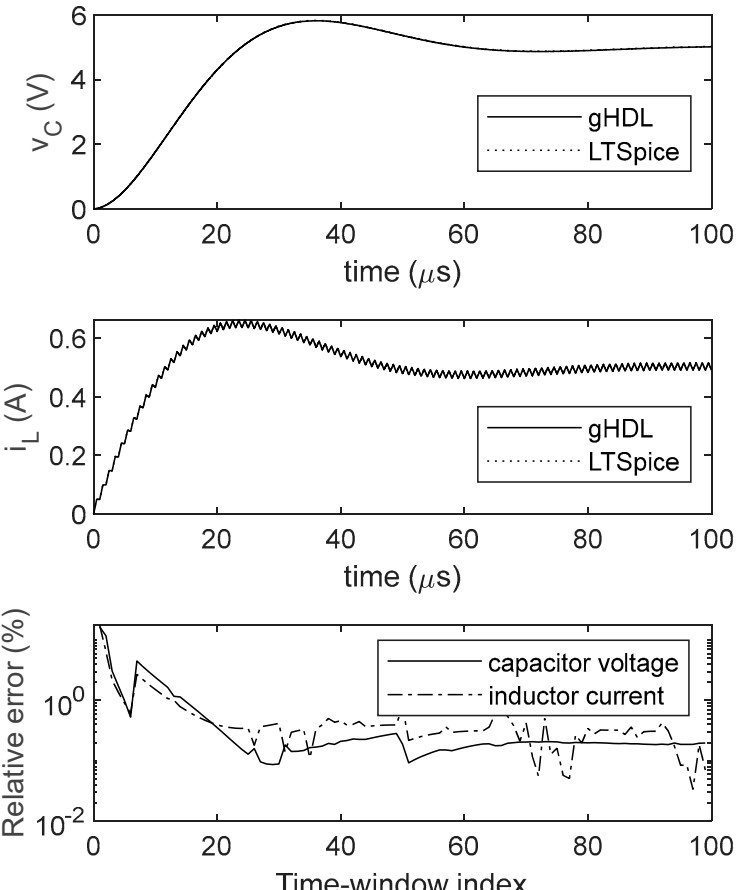

**Figure 3.** Ideal buck power stage waveforms. The first and second plots are the capacitor voltage and inductor current waveforms obtained in gHDL and LTSPICE simulation, respectively. The relative error is drawn in the third plot.

## 2.2. C. Synchronous Buck Converter with Parasitic Components

### 2.2.1. Ordinary Differential Equation System Derived from the Circuit Topology

The synchronous buck model with parasitic components is depicted in Figure 4, alongside its operation modes. In this model, we introduced the effect of the equivalent series resistance of the inductor, noted ESR, and the ON resistance of the power MOSFETs, $R_{DS(on)}$.

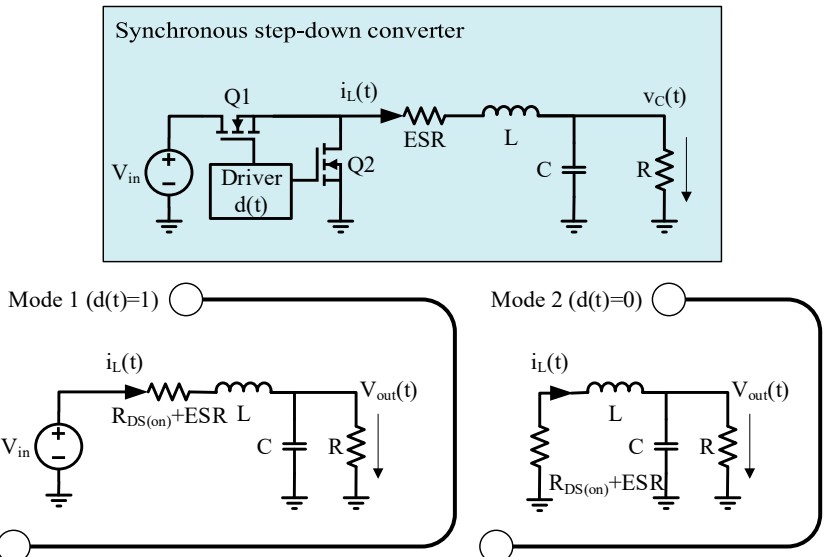

**Figure 4.** Synchronous step-down (buck) converter model with parasitic components and its operating modes.

The differential equation system for the equivalent circuit of Mode 1 and 2 are given in Equations (5) and (6), respectively:

$$\begin{cases} \frac{di_L(t)}{dt} = \frac{1}{L}\left(V_{in} - i_L(t)\left(R_{DS(on)} + ESR\right) - v_C(t)\right) \\ \frac{dV_{out}(t)}{dt} = \frac{1}{C}\left(i_L(t) - \frac{v_C(t)}{R}\right) \end{cases} \tag{5}$$

$$\begin{cases} \frac{di_L(t)}{dt} = -\frac{1}{L}\left(i_L(t)\left(R_{DS(on)} + ESR\right) + v_C(t)\right) \\ \frac{dV_{out}(t)}{dt} = \frac{1}{C}\left(i_L(t) - \frac{v_C(t)}{R}\right) \end{cases} \tag{6}$$

Equation systems (5) and (6) are merged considering the PWM control signal $d(t)$. The resulting ODE is:

$$\begin{cases} \frac{di_L(t)}{dt} = \frac{1}{L}\left[\begin{array}{c} d(t)V_{in} - \\ -i_L(t)\left(R_{DS(on)} + ESR\right) \\ -v_C(t) \end{array}\right] \\ \frac{dv_C(t)}{dt} = \frac{1}{C}\left[i_L(t) - \frac{v_C(t)}{R}\right] \end{cases} \tag{7}$$

### 2.2.2. Numerical Integration with Euler's Method

Applying Euler's method to solve ODE in (7) and considering $i_L(t_0) = 0\ A$ and $v_C(t_0) = 0\ V$ as initial conditions, the iterative process in (8) is derived:

$$\begin{cases} i_L(t_0) = 0, v_C(t_0) = 0 \\ i_L(t_{n+1}) = i_L(t_n) + \left(d(t_n)V_{in} - i_L(t_n)\left(R_{DS(on)} + ESR\right) - v_C(t_n)\right)\frac{\Delta t}{L} \\ v_C(t_{n+1}) = v_C(t_n) - \left(i_L(t_n) - \frac{v_C(t_n)}{R}\right)\frac{\Delta t}{C} \end{cases} \tag{8}$$

### 2.2.3. VHDL Implementation of Euler's Method

The algorithm for computing Equation (8) is very similar to the one given for the ideal synchronous buck converter power stage in Algorithm 1. The sole difference is in line 3: the computation of the inductor current $i_L$ should be modified into

"$i_L$_new $\leftarrow i_L + (d * V_{in} - i_L*(R_{DS(on)} + ESR) - v_C) * \Delta t/L$".

#### 2.2.4. Validation (Optional)

A VHDL module implementing Equation (8) was simulated in gHDL, while its "analog" counterpart was analyzed in LTSPICE. Parasitic components are listed in Table 2. $R_{DS(on)}$, the on-resistance of the power MOSFET, is of 10 mΩ; thus, its effect is present in the simulation results, presented in Figure 5. The first and second plots show the capacitor voltage, $v_C(t)$, and the inductor current, $i_L(t)$. The relative errors are drawn in the third plot. In the transient state of the converter, the relative error peaks at ~10%, while in steady-state operation it is less than 1%.

**Table 2.** Parasitic component values.

| Input | Value |
|---|---|
| $R_{DS(on)}$ | 10 mΩ |
| ESR | 10 mΩ |

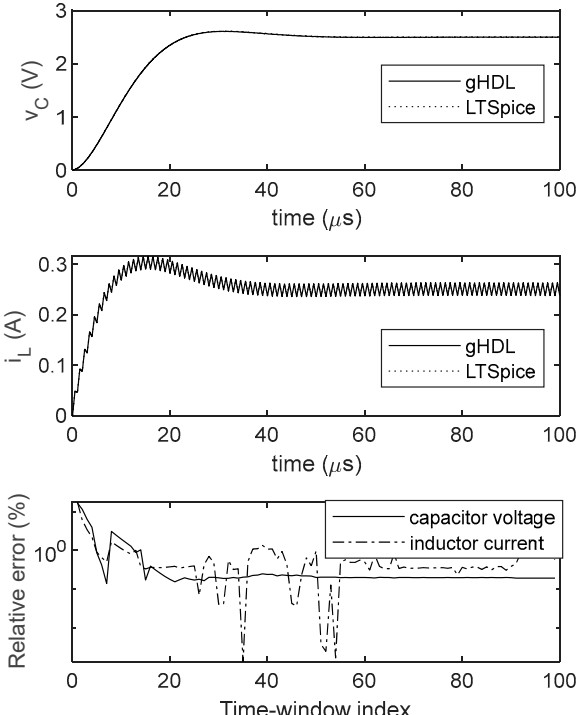

**Figure 5.** Synchronous step-down (buck) converter model with parasitic components and its waveforms. The first and second plots are the capacitor voltage, respectively, and inductor current waveforms obtained in gHDL and LTSPICE simulation. The relative error is drawn in the third plot.

#### 2.3. Ideal Asynchronous Buck

The asynchronous buck converter model was implemented in a previous work [2]; thus, only a brief review is given in this section. The buck converter model is depicted in Figure 6, alongside the iterative process to solve its ODE system in Equation (9).

$$\begin{cases} i_L(t_0) = 0, \ v_C(t_0) = 0 \\ i_L(t_{n+1}) = i_L(t_n) + (d(t_n)V_{in} - d'(t_n)V_D(t_n) + v_C(t_n))\frac{\Delta t}{L} \\ v_C(t_{n+1}) = v_C(t_n) + \left(i_L(t_n) + \frac{v_C(t_n)}{R}\right)\frac{\Delta t}{C} \\ V_D(t_{n+1}) = NV_t\log\left(\frac{i_L(t_n)}{I_S} + 1\right) \end{cases} \quad (9)$$

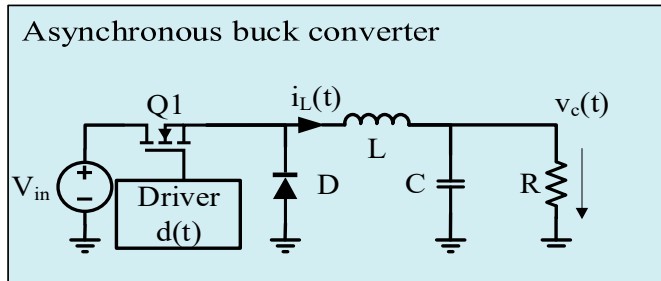

**Figure 6.** Asynchronous buck power stage and the iterative process resulting the inductor current and the capacitor voltage [2].

In the previous section (synchronous buck), the low side switch was modeled as an ideal switch. In this asynchronous buck model, the forward voltage drop on the diode $V_D$ is introduced:

$$V_D = NV_t\log\left(\frac{I_D}{I_S} + 1\right) \tag{10}$$

where $N$ is the ideality factor of the diode, $V_t$ is the thermal voltage at room temperature, $I_s$ is the saturation current, and $I_D$ is the current through the diode, which is equal to $i_L$ in this case.

The comparison of the gHDL and LTSPICE simulation was carried out in [2]. In the transient state of the converter, peak relative error was approximately 15%, and it was decaying as the converter entered a steady state. Once the converter reached a steady state, the relative error reaches 1%.

## 3. VHDL Code Optimization for Hardware Emulation

Power stage emulation for hardware-in-the-loop [18] development can be achieved. The VHDL modules implementing the above-presented buck converter topologies (i.e., see Appendix A) could be synthesized, but, in general, floating-point operations are not very well handled by synthesis tools. Moreover, pipelining is recommended to achieve good timing closure.

Depending on the target technology/device, a few optimizations can be carried out. For example, in FPGAs, the floating-point operations can be carried out by dedicated hardware, as each FPGA vendor has support for arithmetic operations. Our buck power stage emulator was implemented on a 7th Series Spartan FGPA by Xilinx, 7s25csga225-1, using the Vivado design suite.

Dedicated floating-point hardware for multiply, divide and add/subtract operations was generated using Vivado's IP Catalog. Then, the modified VHDL module was elaborated and synthesized. The result of the synthesis is listed in Table 3.

**Table 3.** Resource utilization of the buck power stage emulator for a 7s25csga225-1 device.

| Resource | Used | Available | Utilization |
|---|---|---|---|
| LUT as Logic | 64 | 14,600 | 0.44% |
| Register as Flip Flop | 64 | 29,200 | 0.22% |
| Bonded IOB | 130 | 150 | 86.67% |
| Clocking resources | 1 | 32 | 3.13% |

## 4. PID Controller Implementation Based on Hardware Emulation

### 4.1. Manual Tuning of a Digital PID Controller

Imagine the following use case: one desires to implement a digital control algorithm in an HDL language for a given power stage, but the power stage is not yet manufactured. Therefore, only its design parameters are known: topology (synchronous or asynchronous),



inductance, capacitor, component parasitic (diode serial resistance, forward voltage, etc., or $R_{DS(on)}$ of the MOSFET transistors, ESR of inductance and capacitor less) are known. Generally, the first step is modeling. As the power stage is modeled as an analog circuit, but the control loop is sought to be implemented with digital technology, it is difficult to choose an adequate modeling environment. A first choice is to use a system modeling environment, such as MATLAB/Simulink [19]. This is a powerful tool that (i) can be interfaced with HDL simulation tools, (ii) allows the modeling of active and passive electrical components, and (iii) has support for many signal processing and conditioning components. Another way is to use a mixed-signal design flow from an EDA vendor.

Mixed-signal design environments usually can simulate electrical circuits and HDL code together. Both MATLAB and mixed-signal design flows are closed source, expensive software tools.

As pointed out in previous sections, the buck power stage can be modeled using any HDL languages, and event-driven logic simulators yield accurate results. Thus, we propose to carry out the modeling of both analog and digitals sections in an event-driven logic simulator. A PID controller was designed to demonstrate the ease of use, and manual tuning was carried out to find out the PID controller parameters.

A popular adaptive algorithm for power converter control employs PID control [20]. Figure 7 shows the diagram of a buck converter with a PID controller. The PID regulator compares the value of the voltage measured at the output of the $v_{out}(t)$ power stage, digitized through a digital-to-analog converter, with the reference value specified $V_{ref}$. The converter works with a sampling frequency equal to 1 MHz. 10 samples are averaged, so the input of the PID controller is $v_{out}[n]$ at the frequency of 100 kHz. Then, the error $e[n]$ is processed to calculate the duty cycle of the PWM modulated signal $d(t)$ controlling the power stage. The error signal is processed by the proportional, integrative and derivative blocks, resulting in $p[n]$, $i[n]$ and $d[n]$ signals, respectively.

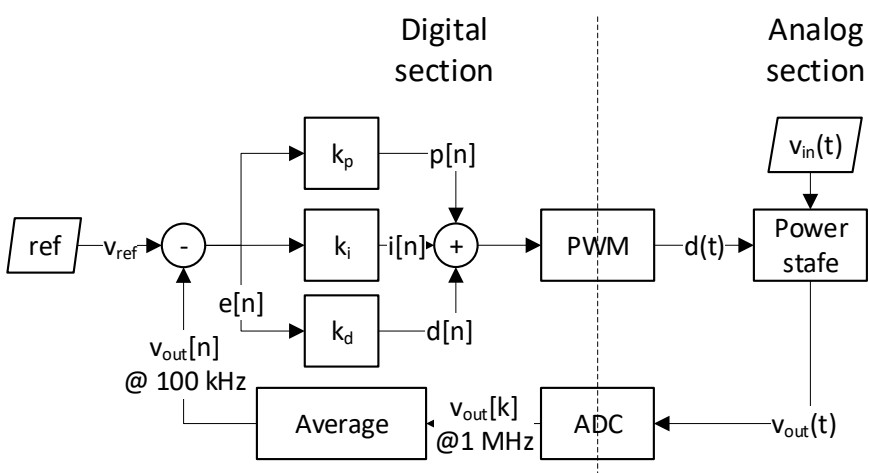

**Figure 7.** PID controller and a buck converter's power stage simulation.

The "proportional" block computes the proportional component of the setting, having the computing formula:

$$p[n] = k_d \cdot e[n] \tag{11}$$

The integration performed by the "integration" block is represented by the equation:

$$i[n] = k_i \cdot (acc[n] + e[n]) \tag{12}$$

where $acc[n]$ represents the accumulated value of the integrative component and is calculated by:

$$acc[n+1] = acc[n] + e[n] \tag{13}$$

The "derivative" block computes the derivative part by implementing the relation:

$$d[n+1] = k_d \cdot (e[n] - e[n-1]) \tag{14}$$

The components in Figure 7 were described in VHDL. The parameters of the power stage are summarized in Table 4. In the analog section, the numerical values were represented using floating-point representation. Note that this section is not intended to be later synthesized. The boundaries between the analog and digital sections are marked by the pulse width modulator (noted PWM in Figure 7) and analog-digital converter (ADC). The sampling frequency of the converter is $F_{sampling}$, while the switching frequency of the pulse width modulator is $F_{swithcing}$. The digital section can use fixed-point numerical representation instead of power and aria-hungry floating-point presentation, as the ADC has a finite resolution anyhow. In the actual VHDL description, we used 32 bits wide signals, 22 bits for the integer part and 10 bits for the fractional part (basically, a scaling factor of $2^{10}$ was applied). The PID controller was manually tuned, obtaining the values in Table 5.

**Table 4.** Buck power stage parameters.

| Input | Value |
|:---:|:---:|
| $L$—inductor | 22 μH |
| $C$—capacitance | 440 μF |
| $V_{in}$ | 10 V |
| $V_{out}$ | 3.3 V |
| $F_{swithcing}$ | 100 kHz |
| $F_{sampling}$ | 1 MHz |

**Table 5.** PID Controller Parameters.

| Input | Scaled Value | Unscaled Value |
|:---|:---:|:---:|
| Proportional weight, Kp | 512 | 0.5 |
| Integrative weight, Ki | 0.009 | 9 |
| Derivative weight, Kd | 1228 | 1.2 |

Several analyses were carried out using the gHDL logic simulation: buck startup/soft-startup, load regulation and line regulation. These results are presented in Figure 8. Figure 8a shows the response of the PID converter at start-up for an input voltage of 12 V and a load of 2 Ohm. If the "soft start" function is not activated, the voltage at the converter output has an overshoot of 1.16 V and a stabilization time of approximately 4 ms. With the "soft start" function active (by slowly varying the reference voltage $V_{ref}$), the voltage increase is negligible.

In Figure 8b, the result of the "load regulation" analysis is presented. At 5 ms, the load changes from 2 Ohm to 4 Ohm, and at 20 ms, it changes to 2 Ohm. The supply voltage is 12 V. The undershoot and overshoot at the load variation are less than ±3% of the desired voltage value of 3.3 V.

Figure 8c presents the result of the "line regulation" analysis with a constant load of 2 Ohm. If the supply voltage changes at 4 ms, the $V_{in}$ supply voltage changes from 12 V to 16 V, and at 8 ms, it changes back to 12 V. The overshoot at the variation of the supply voltage is 475 mV, and the undershoot increase is 273 mV.

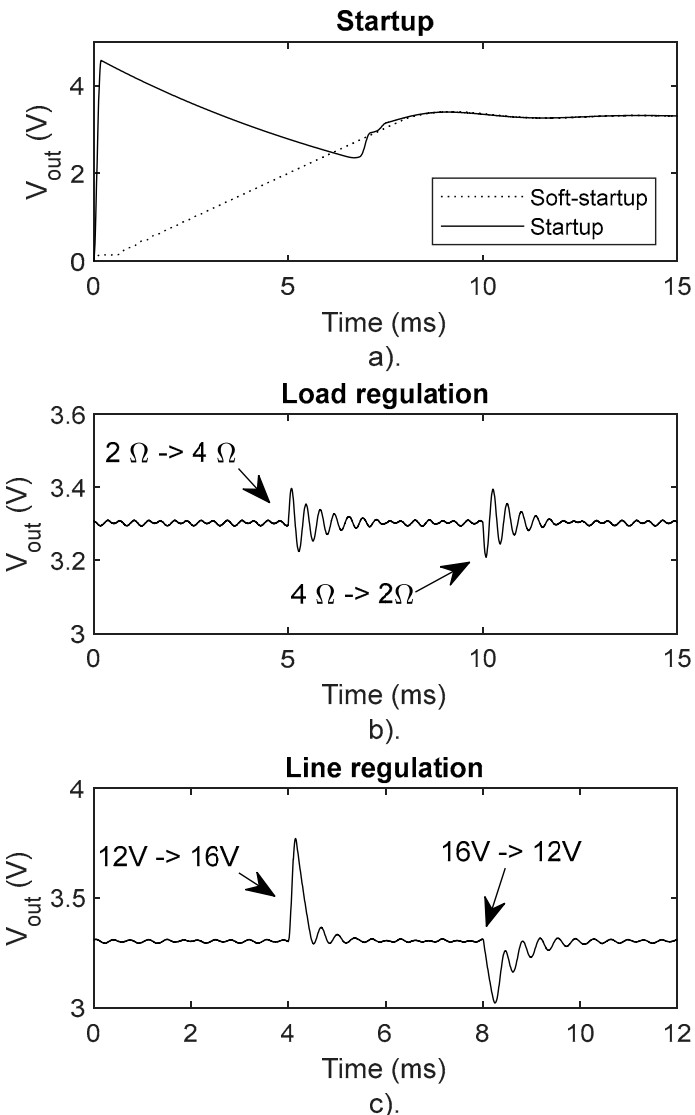

**Figure 8.** Simulation results: (**a**) power stage startup with/without soft-startup; (**b**) load regulation analysis; (**c**) line regulation analysis.

### 4.2. PID Controller Implementation on FPGA

Let us imagine another use case: one desires to implement a digital control algorithm for an off-the-shelf (COTS) buck power stage, i.e., TI's Digital Power Buck Converter BoosterPack [21] using a FPGA. Nowadays, even FPGAs incorporated ADC. Thus, it is recommended to select one with this feature. Our choice was a low-cost evaluation board CMOD-S7 [22] equipped with 7th Series Spartan FGPA by Xilinx, 7s25csga225-1. A hardware emulator modeling the Digital Power Buck Converter BoosterPack was developed as a first step of the design. For the sake of this example, let us suppose that the hardware emulator was necessary because a long shipping time of the actual power stage hardware was too long, so in its absence, development had to be carried out without it.

The experimental setup is depicted in Figure 9. The BoosterPack consists of a synchronous buck power stage and other complementary circuits for: (i) driving the high and low side switches $Q_1$ and $Q_2$; (ii) measuring the inductor current $i_L$; and (iii) creating a feedback voltage $V_{fb}$ with the use of a resistive voltage divider.

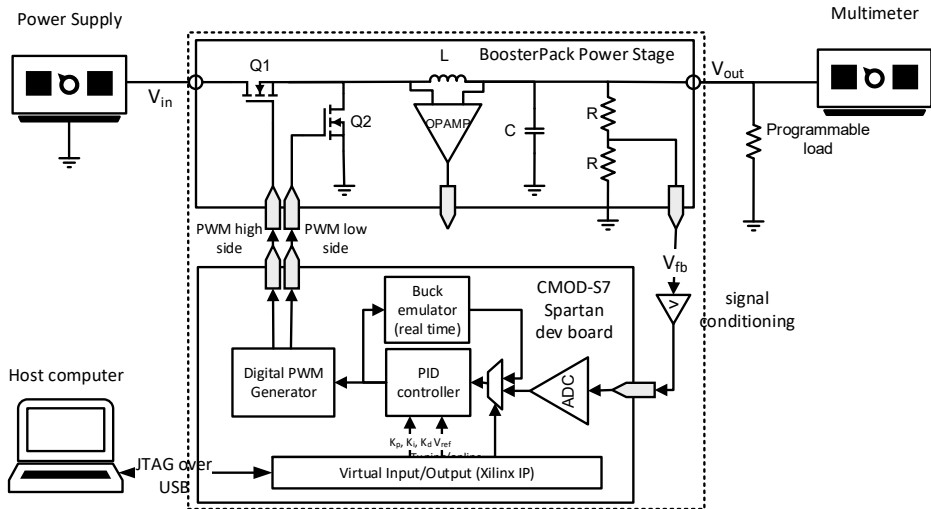

**Figure 9.** Experimental setup for buck converter characterization and measurements.

The CMOD-S7 dev board is used to implement the digital control loop consisting of: (i) an ADC converter fitted in FPGA SoC; (ii) the digital PID controller; (iii) a digital PWM generator; (iv) a Virtual Input/Output (VIO) Interface—a Xilinx intellectual property (IP) [23]—connected to a host computer by a JTAG over USB port; and (v) the power stage emulator. The feedback voltage $V_{fb}$ is lowpass filtered to prevent aliasing effects. In addition to filtering a voltage limiting is inserted as the dynamic range of the internal ADC is 1 V. A programmable load is connected to the output of the power stage, allowing the simulation of a changing load. A programmable voltage supply is used to sweep $V_{in}$, the input voltage, and a multimeter is connected to $V_{out}$, the voltage output of the power stage. The host computer is running the VIO application, thus the user can programmatically set the reference voltage $V_{ref}$. This way the user can effectively set the desired output voltage. Moreover, the VIO interface allows the user to set the PID controller's coefficients $K_p$, $K_i$ and $K_d$., and thus the PID controller can be configured. The VIO facilitates the selection of two paths with the use of a multiplexer: a controller tuning path, where the hardware emulator takes the place of the actual power stage, and an online operation path, when the actual power stage is connected to the control loop.

When the actual power stage arrived, the setup in Figure 10 was assembled. With the PID controller already tuned, the user can switch to online operation mode. The setup is ready, and measurements can be conducted. The COTS components are highlighted and numbered in the figure.

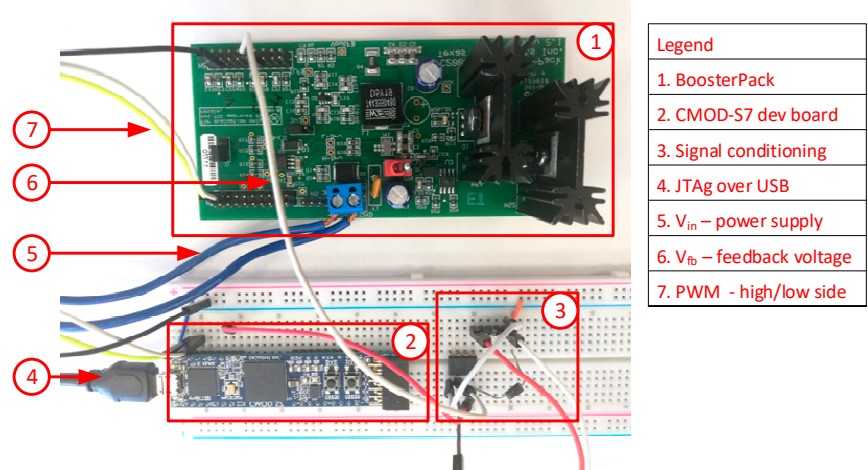

**Figure 10.** PID controller on a CMOS-S7 module and TI's BoosterPack buck power stage.

In Figure 11, we present the result of three tests: we varied the reference voltage, $V_{ref}$ to 1.2 V, 1.8 V and 3.3 V, while the input voltage, $V_{in}$, was swept between 7 V to 14 V with a step of 0.1 V. The output voltage of the converter was as required, and the voltage ripple was measured and plotted. The same measurement was carried out with the power stage's reference digital voltage mode control. As a conclusion, one can state that the designed PID controller has slightly higher ripple then a standard voltage mode control, but the order of magnitude is the same.

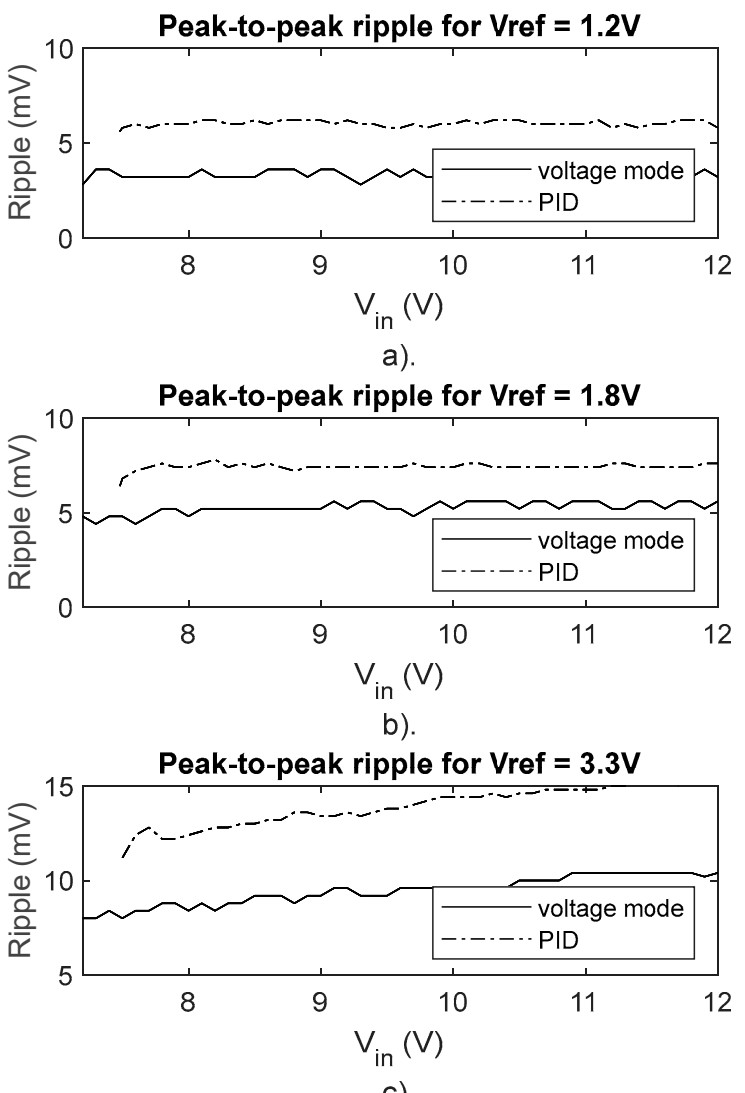

**Figure 11.** Voltage ripple for: (**a**) $V_{ref}$ = 1.2 V, (**b**) $V_{ref}$ = 1.8 V, (**c**) $V_{ref}$ = 3.3 V.

## 5. Results and Discussion

In the present paper, we presented a methodology to accurately model the behavior of buck converter power stages using VHDL. The obtained VHDL module can be synthesized, yielding a hardware emulator. In Section II, three buck power stage topology was described in VHDL and validated through comparison with LTSPICE electrical models: an ideal synchronous buck (with ideal switches), a synchronous buck (with parasitic components, as ESR and on-resistance of the switches) and an asynchronous buck (modelling the diodes forward voltage and ideality factor).

In Figure 3 the ideal buck power stage waveforms were presented. The first and second plots are the capacitor voltage and the inductor current waveforms obtained in gHDL and LTSPICE simulation. In the transient state, the relative errors of the waveforms

are ~10%, but in the steady-state operation, it is less than 1%. The waveforms of a buck converter with parasitic components are presented in Figure 5. In the transient state of the converter the relative error peaks at ~10%, while in steady-state operation, it is less than 1%. Finally, the asynchronous buck converter power stage was analyzed in [2]. In the transient state of the converter, peak relative error was approximately 15%, and it was decaying as the converter entered a steady-state. Once the converter reached a steady-state, the relative error approximately 1%

In Section 3, the hardware emulator was implemented in a 7th Series Spartan FGPA. The implementation makes use of the FPGA vendor's floating-point operation support. The resource utilization of the emulator of a synchronous buck power stage with parasitic components is reported in Table 3.

The hardware emulators analyzed in the present work are compared in terms of target technology, computation kernel, numerical representation, time resolution and availability (see Table 6). The proposed methodology is compared to existing ones, given in refs. [9–12]. The de-facto choice for target technology is the FPGA. The computation kernel ranges from ODE solvers to average state space models. ODE solvers are the preferred ones as they yield better accuracy. Most hardware emulators are custom made, tailored for the given application, but a few commercial emulators are also available. The time resolution is increasing with the advances of the FPGA technology.

**Table 6.** Hardware emulator comparison.

|  | **Proposed** | **Ref. [9]** | **Ref. [10]** | **Ref. [11]** | **Ref. [12]** |
|---|---|---|---|---|---|
| Target Technology | FPGA | FPGA | MPU | FPGA | FPGA |
| Kernel | ODE solver | ODE solver | Average State Space Solver | ODE solver | N/A |
| Numerical Representation | Fixed Point | Floating Point | 32 bits | N/A | N/A |
| Time Resolution ($\Delta t$) | 10 ns | 20 ns | Order of μs | 50 ns | Order of μs |
| Availability | Custom | Custom | Custom | Custom | Commercial |

Section 4 presents the PID controller design based on hardware emulation. The first phase focuses on the utility of VHDL description of the power stage. Noteworthy to mention is that computer modeling of the converter circuit is hardened by its mixed signal nature, as it contains both analog and digital components. The designer is forced to use either an expensive and resource-hungry mixed signal design environment or a system level modeling environment. In our approach, the analog section is discretized, thus the designer can use digital hardware design environment suited for FPGAs. The VHDL description allows the accurate simulation of the buck power stage in an event-driven simulator. This enabled the tuning of the PID controller manually. The event-driven simulator was used to perform a few tests: buck startup/soft-startup, load regulation and line regulation. They are presented in Figure 8 and commented on in Section 4.1. In the second phase, the power stage was emulated on FPGA, and a PID controller was devised. The performance of the obtained PID controller was compared to a standard voltage mode control. As output voltages for both controllers were required, the ripples in the steady-state of the two control algorithms were evaluated. In Figure 11, we presented the ripple for three reference voltages, 1.2 V, 1.8 V and 3.3 V, while varying the input voltage in an extensive range, from 7 V to 14 V. The two algorithms have comparable ripple amplitudes.

Further work will comprise the next ideas: the development of an EDA tool that automatically generates the necessary FPGA configurations of the desired power stage, or more generally, the desired ODE extending the hardware emulation to other SMPS topologies, such as multiphase buck converter [24], three-level flying capacitor buck converter [25]

and other types of DC/DC converters (boost, buck-boost); the hardware emulation of AC/DC and DC/AC converters shall be considered in the close future; the educational use of hardware emulators shall be considered.

## 6. Conclusions

This paper presents a generalized methodology for designing FPGA-based emulation hardware of several step-down (buck) converter power stages. The presented methodology is used in the development of a digital control loop based on PID control. The hardware emulator allows the off-line tuning of the PID controller and on-line operation of the digital control loop with an actual power stage.

The hardware emulation—in other terminology, the real-time simulation—of the power stage is based on solving the ordinary differential equation system extracted from the power stage topology. The two most important emulated quantities are the capacitor voltage and the inductor current of a power stage.

Section 2 guides the reader through the process of delivering the emulation hardware of a power stage. First, the power stage is described using an ordinary differential equation system; second, the ordinary differential equation system is solved using Euler's method, thus an accurate time-domain model is obtained; next, this time-domain model can be described using either general-purpose programming language (MATLAB, C, etc.) or hardware description language (VHDL, Verilog, etc.). As a result, the emulator has been created; validation of the emulator may be carried out by comparing it to SPICE transient simulations; finally, the validated emulator can be implemented on the preferred target technology: either in a general-purpose processor or a field programmable gate array.

In this paper, several buck power stage topologies were emulated: (i) ideal synchronous buck, (ii) synchronous buck converter with parasitic (direct current resistance of the inductance, the equivalent series resistance of the capacitor, on-resistance of MOSFET switches), and (iii) ideal asynchronous buck.

The VHDL modules implementing the above-mentioned buck converter topologies are synthesized to achieve FPGA hardware emulation. The numerical representation used in the emulator is an important aspect. In general, floating-point operations are not very well handled by synthesis tools. Our implementation uses fixed-point representation instead of power and aria-hungry floating-point. The ADC has a finite resolution anyhow. In the actual VHDL description, we used 32 bits wide signals, 22 bits for the integer part and 10 bits for the fractional part.

The FPGA emulator was used in the development of a digital PID controller. Our experimental setup facilitates the selection of two hardware loops: an off-line loop is used to tune the PID controller while the power stage is emulated; an online loop takes the place of the actual power stage; and an online operation loop, when the manually tuned PID controller is connected to the actual power stage.

**Author Contributions:** Conceptualization, B.S.K.; Methodology, B.S.K.; Formal analysis, C.C. and I.-A.I.; Writing—original draft, B.S.K.; Writing—review & editing, C.-A.F. and M.N. All authors have read and agreed to the published version of the manuscript.

**Funding:** This work was co-funded by the European Regional Development Fund through the Operational Program "Competitiveness" POC -A1.2.3-G-2015, project P_40_437, contract 19/01.09.2016, SMIS code 105742.

**Data Availability Statement:** Not applicable.

**Acknowledgments:** The authors express their gratitude for the anonymous reviewers. Their valuable observations were a great help in improving the quality and content of this paper.

**Conflicts of Interest:** The authors declare no conflict of interest.

## Appendix A

The VHDL module presented in Figure A1 was written for buck power stage simulation. The buck power stage parameters are passed to the module as global (generic) parameters. The module has an input signal, the PWM control signal *d(t)*, two output signals, the inductor current, and the capacitor/output voltage.

```vhdl
entity ideal_syncronious_buck is
    generic ( L : real;
              C : real;
              deltaT : time);
    port ( Vin : in real := 0.0;
           duty : in real := 0.0;
           Rload : in real := 1.0;
           Vout : out real := 0.0;
           iL : out real := 0.0);
end entity;

architecture euler of ideal_syncronious_buck is
        function time2real (t : in time) return real is
        begin
            return real(deltaT / 1 ns) * 1.0e-9;
        end function;
begin
    process begin
        iL <= 0.0;
        Vout <= 0.0;
        while true loop
            wait for deltaT;
            iL <= iL + (duty * Vin - Vout) * time2real(deltaT) / L ;
            Vout <= Vout + (iL - Vout/Rload) * time2real(deltaT) / C;
        end loop;
    end process;
end architecture;
```

**Figure A1.** VHDL module of the ideal synchronous buck power stage.

## Appendix B

The buck power stage was modeled in LTSPICE (Figure A2). The switching components were modeled with ideal switches, including on and off resistances. The PWM control signal with a 50% duty cycle was generated using Pulse voltage sources.

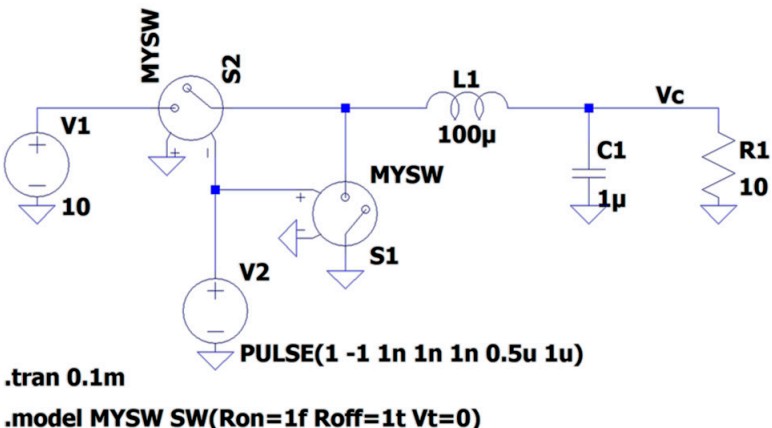

**Figure A2.** LTSPICE model of the buck power stage.

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
