# Peer review of "Hardware Emulation of Step-Down Converter Power Stages for Digital Control Design"

_electronics, doi:10.3390/electronics12061328_

Round 1
Reviewer 1 Report
1. Abstract is too specific. Please highlight the novelty compared with previous studies and summarize your contribution.
2. The operation frequency (not the step size) should be provided in Table 1. It seriously influences the result.
3. Based on Table 1 and Table 2, if the Rds(on) is 10Ω, it is the same value as the R-load. It means the voltage drop across the MOSFET Vds is the same as Vout, resulting in a lower 50% efficiency. What is your comment on this problem?
4. Why are the parameters for Table 4 and Table 1 not matched?
5. Page 11, line 308, the author mentioned that the PID controller has a higher ripple than a standard voltage mode control. Figure 11 shows the data in the opposite situation. Additionally, the kp, ki, and kd can change the ripple successively with varied values. It means little.
6. How to prove the proposed model is accurate? Any comparison with another precise model/algorithm? There is no evidence now.
Author Response
Dear Reviewer 1!
First, let me express our appreciation for your valuable comments. Hopefully, the answers will be objective and convincing. Note that, the major changes made are highlighted in the article with red color text.
Comment 1 ("Abstract is too specific...")
Response: As all of the reviewers raised concerns about the Abstract and Introduction section, the Abstract was rewritten and the Introduction section was completed. Because Reviewer 3 asked to make the Abstract short and precise, we fitted his request in the revised paper. In the Introduction section we introduced new paragraphs with almost 10 new and quality references. Also in the Introduction, the original contributions of the article are indicated. To improve the structure of the paper, we introduced subsections in Section 2.1. and 2.2.
Comment 2 ("The operation frequency (not the step size) should be provided in Table 1...")
Response: we inserted the switching frequency of the power stage, it is 1MHz.
Comment 3 ("Based on Table 1 and Table 2, if the Rds(on) is 10Ω, it is the same value as the R-load...")
Response: our intention is to exemplify the process of obtaining the a synthesizable VHDL code, with other words the FPGA based hardware emulator. The parameters of the power stage in Table 1 are just example values, there were not obtained after dimensioning the stage for a given specification.
Comment 4 ("Why are the parameters for Table 4 and Table 1 not matched?")
Response: the power stage parameters in Table 1 are just example values, while parameters in Table 4 belongs to an existing power stage. We considered changing the values in Table 1 for the revision, but it will affect the results presented in Figure 4. Also Appendix A and B should be corrected accordingly. If necessary, we can make this change, but it will alter a big portion of the paper.
Comment 5 ("Page 11, line 308, the author mentioned that the PID controller has a higher ripple than a standard voltage mode control...")
Response: indeed, the voltage mode controller marginally outperforms our PID controller. Still, we hope the reviewers find valuable confirmation in our conducted experiments. Also, we wish to stress that this article gives a good insight in how to carry out hardware emulation staring from the circuit topology resulting in an FPGA hardware emulator.
Comment 6 ("How to prove the proposed model is accurate?...")
Response: the VHDL module is a bit accurate model of the synthesized hardware emulator. This means that the hardware emulator is functioning precisely as the VHDL module. The VHDL module is validated by comparing it to an electrical model carried out in LTSpice. This comparison is illustrated for the ideal buck power stage in Figure 4. The relative error between the VHDL model and the LTspice model are depicted in plot 3. We can conclude that in the case of the ideal buck, the relative error is less then 1% when steady state operation of the converter is reached. The same comparison is carried out for a realistic buck converter in Figure 6.
Reviewer 2 Report
The paper is interesting but needs to be improved:
1.. The bibliographic references must be improved and updated, several recent papers from 2017 to 2023 must be cited. In his paper it is observed that more than half of the cited papers are very old.
2.. It is necessary to make a broader review of the state of the art with new papers.
3.. The formulation of the problem and the objectives of the research are not clear.
4.. A list of contributions of the paper should be made, please add it at the end of the introduction.
5.. Improve the quality of figure 1 and 13 in the other results figures if you want you can use colors for example in figure 7, 11.
6.. In figure 9, where load regulation is made, it is necessary to draw the current signal in the load
7.. After a number you must make a space and then put the units, for example looking at the (Table 1,4) 10V must be changed to 10 V (leaving a space), make these corrections throughout the entire document.
8.. In the results make a change in the reference signal both up and down in the reference signal.
9.. If it is possible in the results make a change in the Vin to see the behavior of the controlled signal Vout
10.. The conclusions should be improved, with respect to the objectives and contributions of the paper.
Author Response
Dear Reviewer 2!
First, let me express our appreciation for your valuable comments. Hopefully, the answers will be objective and convincing. Note that, the major changes made are highlighted in the article with red color text.
Comment 1 ("The bibliographic references must be improved and updated...")
Response: As all of the reviewers raised concerns about the Abstract and Introduction section, the Abstract was rewritten and the Introduction section was completed. Because Reviewer 3 asked to make the Abstract short and precise, we fitted his request in the revised paper.
Comment 2 ("It is necessary to make a broader review of the state of the art..." )
Response: In the Introduction section we introduced new paragraphs with almost 10 new and quality references (see lines 57-73 in Introduction section).
Comment 3 ("The formulation of the problem and the objectives of the research are not clear")
Response: in this paper we intend to give a precise methodology of deriving a hardware emulator starting from the circuit topology of a power stage; the methodology is demonstrated for 3 circuit topologies (ideal synchroniuous buck, realistic synchroniuous buck and asyncronious buck converter); using the hardware emulation a digital PID control loop was developed for FPGA.
Comment 4 ("A list of contributions of the paper should be made")
Response: In the Introduction, the original contributions of the article are indicated in lines 74-108.
Comment 5 ("Improve the quality of figure 1 and 13 in the other results figures if you want you can use colors for example in figure 7, 11)
Response: We changed the font in Figure 1 to Arial; Figure 13 (renumbered to Figure B1, as required by the journal tamplete) is larger. We colored result figures, now Figure 4 and 6, but the two results (VHDL and LTSPICE simulations) are overlapping, just one color is visible. We left these figures in grayscale.
Comment 6 and 9 ("In figure 9, where load regulation is made, it is necessary to draw the current signal in the load..."; ("In the results make a change in the reference signal both up and down...")
Response: the timeframe provided for the review (10 days) is not enough to complete this request. We engage ourself to respond for this objection in a "minor revision"
Comment 7 ("After a number you must make a space...")
Response: Done.
Comment 9: ("make a change in the Vin to see the behavior of the controlled signal Vout")
Response: This experiment was conducted, the results are shown in Fig. 11c (line regulation); vin is changed from 12 V to 16 V and back;
Comment 10 ("The conclusions should be improved, with respect to the objectives and contributions of the paper")
Response: we considerably improved the Results and Discussions section, a comparison is given with state-of-the-art solutions (see lines 375-379; 387-401)
Reviewer 3 Report
Dear authors,
Please see the attached file for my review of your manuscript.
Please note that all comments are of qualitative nature.

Author Response
Dear Reviewer 3!
First, let me express our appreciation for your valuable comments. Hopefully, the answers will be objective and convincing. Note that, the major changes made are highlighted in the article with red color text.
Comment 1 ("The abstract of the manuscript is very poorly written and too long...")
Response: As all of the reviewers raised concerns about the Abstract and Introduction section, the Abstract was rewritten and the Introduction section was completed. As you asked to make the Abstract short and precise, we fitted your request in the revised paper.
Comment 2 ("... Introduction section should be extended to mention novel advances compared to other literature solutions...")
Response: In the Introduction section we introduced new paragraphs with almost 10 new and quality references (see lines 57-73). Also in the Introduction, the original contributions of the article are indicated (see lines 74-108). To improve the structure of the paper, we introduced subsections in Section 2.1. and 2.2.
Comment 3 ("In terms of editing, figures, caption: ...")
Response: all the comments of the reviewer were took into account and it was corrected
Comment 4 ("... a comparative data table should be added compared to other solutions that can be found in the literature...")
In section 5. Results and Discussions a comparison of existing hardware emulators were compared, see lines 384-401.
Comment 5 ("Conclusions section needs to be extended and rewritten...")
Response: further work was added in section 5. Results and Discussions, see lines 411-418.
Comment 6 ("The reference list is quite short ...")
Response: 9 new quality references were introduced, reference 9-12 are compared with the existing solution.
Response:
Round 2
Reviewer 1 Report
A Rds(on) with 10Ω is quite rare, which even cannot be found in real power transistors. Please consider this problem.
Author Response
Dear Reviewer 1!
Comment 1 ("A Rds(on) with 10Ω is quite rare, which even cannot be found in real power transistors. Please consider this problem.")
Response: Indeed 10Ω is quite high for RDS-on. We changed it to 10 mΩ and rerun the simulations. It's effect doesn't show up quantitatively. The text was also updated in Table 1. and line 226-227.
Thank you again for your valuable comments, I really think it helped a lot to significantly improve the quality of this paper.
Reviewer 2 Report
Thank you very much for the corrections.
Author Response
Thank you again for your valuable comments, I really think it helped a lot to significantly improve the quality of this paper.
Reviewer 3 Report
Dear authors,
Thank you for your careful revision. However, some minor edits are still necessary before publishing.
Please see the attached file.

Author Response
Dear Reviewer 3!
Thank you again for your valuable comments, I honestly think it helped a lot to significantly improve the quality of this paper. Our responses as follows:
Comment 1 ("Even though this was not mentioned in the previous review round, Figure 10..")
Response: A new figure (currently Figure 10) and description of the experimental setup is given. Please find it in lines 349-376.
Comment 2 ("The Conclusion Section should be improved for clarity and extended to reflect possible future improvements..."):
Response: Future improvements were included in Section 5. Results and Discussion (see lines 436-442). The conclusion section was also rewritten. See modifications in lines 444-448, 454-461, 466-472 and 473-477
Comment 3 (typo in Appendix A):
Response: Corrected.